# PerPO: Perceptual Preference Optimization via Discriminative Rewarding

## Abstract

This paper presents **Per**ceptual **P**reference **O**ptimization (**PerPO**), a perception alignment method aimed at addressing the visual discrimination challenges in generative pre-trained multimodal large language models (MLLMs). PerPO employs discriminative rewarding and listwise preference optimization to align MLLMs with human visual perception processes. By utilizing the reward as a quantitative margin for ranking, our method effectively bridges generative preference optimization and discriminative empirical risk minimization. PerPO significantly enhances MLLMs' visual discrimination capabilities while maintaining their generative strengths, mitigates image-unconditional reward hacking, and ensures consistent performance across visual tasks. This work marks a crucial step towards more perceptually aligned and versatile MLLMs. We also anticipate that PerPO will inspire the community to reconsider MLLM alignment strategies.

## 1 Introduction

The success of *next token generation* (Radford, 2018; Radford et al., 2019) has reignited the pursuit of artificial general intelligence (AGI). Representative methods (Brown, 2020; Anthropic., 2024) have achieved non-trivial advancements in both creative generation (Zhao et al., 2024b; Azaiz et al., 2024) and logical reasoning (Yang et al., 2023a; Frieder et al., 2023). Recently, they have also demonstrated exceptional multimodal capabilities (Achiam et al., 2023; OpenAI., 2024), achieving remarkable results in various generative visual tasks (Yang et al., 2023b; Wen et al., 2024).

However, visual discrimination tasks have emerged as the Achilles' heel of these multimodal large language models (MLLMs) (Li et al., 2024b; Qu et al., 2024; Liu et al., 2024a). These tasks, which require minimal reasoning and yield deterministic answers—such as "provide the position of the person", as illustrated in Figure 1a—often leave these powerful models quite "nearsighted", or even "blind". Could it be that *generative models fundamentally **struggle with visual discrimination tasks** that are simple for a child?*

Despite efforts (Yu et al., 2023; Wei et al., 2023) to address this issue by incorporating discriminative tasks into generative pre-training, results often remain suboptimal, compromising core linguistic abilities. This paper approaches the problem from an ***alignment*** perspective. We argue that *performance deficiencies in pre-trained models with basic competencies stem primarily from misalignment*. In practice, existing MLLMs lack alignment with perceptual objectives—a fundamental expectation for such models. Recent methods (Sun et al., 2023; Zhao et al., 2023) using Direct Preference Optimization (DPO) (Rafailov et al., 2024) aim for low-hallucination, high-accuracy outputs but often fall into image-unconditional reward hacking (Skalse et al., 2022), a phenomenon where text preferences are optimized without truly engaging with visual input. Consequently, a truly perception-oriented alignment becomes increasingly necessary.

In this paper, we propose a simple yet effective approach: **Per**ceptual **P**reference **O**ptimization (PerPO) via *discriminative rewarding*. Our method aims to align with humans' innate, coarse-to-fine visual perception process: implicitly generating various hypotheses around the objective ground truth, then progressively focusing along the path of increasing rewards towards the optimal hypothesis (Hegdé, 2008). To simulate this process, PerPO extends the wisdom of empirical risk minimization (Pérez-Cruz et al., 2003; Golubev, 2004), initially defining the reward as the negative value of the errors between model predictions relative to the objective ground truth. Figure 1b shows, through a Best-of-N (Charniak & Johnson, 2005) validation, the remarkable consistency

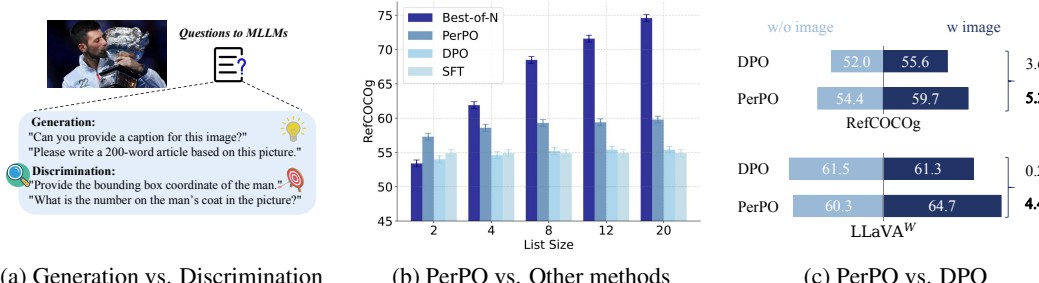

(a) Generation vs. Discrimination     (b) PerPO vs. Other methods     (c) PerPO vs. DPO

Figure 1: (a) Examples of visual generative and discriminative tasks. (b) Performance comparison in RefCOCOg (Mao et al., 2016) with increasing list size for SFT, DPO, PerPO, and Best-of-N. (c) Performance comparison of PerPO and DPO with and without image input across different benchmarks. Notably, PerPO shows a greater performance gap, highlighting a strong reliance on image conditioning.

between this reward and visual discriminative ability, also revealing the untapped discriminative potential within MLLMs.

Centered on such discriminative reward, PerPO first employs a learning-to-rank (Burges et al., 2005) approach for **listwise preference optimization** (Liu et al., 2024d) over an ordered set of all negative samples. Where the negative samples are model-generated responses that deviate from the ground truth, and the "negative" is relative to the discriminative ground truth. This strategy aims to fully exploit the inherent *scalability* of discriminative rewards, enabling efficient learning from diverse negative samples without human annotation. It is also founded on our intuition that ordered sequences of samples, rather than isolated pairs, *can better capture image-conditioned preference patterns*. As Figure 1c confirms, PerPO significantly suppresses optimization toward image-unconditioned reward hacking. Meanwhile, to compensate for the *uncertainty* introduced by preference ranking, we treat the **reward itself as a quantitative margin** for anchoring the ranking. We demonstrate both theoretically and empirically that PerPO effectively combines generative preference optimization with discriminative empirical risk minimization. This ultimately ensures consistent modeling across visual generation and discrimination tasks.

Our contributions are summarized as follows:

1. We highlight, for the first time, the capability dilemma of generative MLLMs in visual discrimination tasks. To address this, we propose PerPO, the first method to align with the human perception process, enhancing both visual discrimination performance and human preference alignment.

2. Technically, we first introduce a scalable discriminative reward that aligns well with both perception and human preferences.

3. Building on this, a listwise approach to preference optimization effectively distills insights from diverse negative samples and mitigates image-unconditional reward hacking.

4. Further, using the reward itself as a margin to anchor uncertainty in ranking is theoretically and experimentally proven to harmonize visual perception and generation.

## 2 PRELIMINARIES

**Best-of-N** sampling (Charniak & Johnson, 2005; Nakano et al., 2021), also known as rejection sampling, involves generating N candidate solutions and selecting the one that scores highest according to a proxy reward. This method leverages the natural *variability* (Renze & Guven, 2024) in LLM responses, effectively finding the best output from a pool of possibilities. By picking the top-scoring candidate, Best-of-N increases the likelihood of identifying the correct answer, enhancing the problem-solving capabilities (Guo et al., 2024) of LLMs and making them more reliable and accurate (Bai et al., 2022).

**Direct Preference Optimization** (DPO) (Rafailov et al., 2024) surpasses Best-of-N by utilizing an *implicit reward* derived from reinforcement learning objectives. DPO employs the LLM for both reward learning and proposal generation, fine-tuning the model to better align with human preferences. This integration improves the model's relevance and quality, pushing the boundaries of LLM performance. Formally, given pairwise preference data $(x, y^+, y^-)$, where $y^+$ is preferred over $y^-$ with respect to prompt $x$, the reward objective is defined as:

$$r(x, y) = \beta \log \frac{\pi_\theta(y|x)}{\pi_{\text{ref}}(y|x)} + Z(q) \tag{1}$$

where $\pi_\theta$ is the model being optimized, $\pi_{\text{ref}}$ is the reference model, $Z(q)$ is a partition function, and $\beta$ is a hyperparameter controlling the deviation between $\pi_\theta$ and $\pi_{\text{ref}}$. By reparameterizing the Bradley-Terry (BT) model (Bradley & Terry, 1952), DPO's objective can be expressed as:

$$\mathcal{L}_{\text{DPO}}(\theta) = -\mathbb{E}_{(x, y^+, y^-) \sim \mathcal{D}}[\log \sigma(\beta(\log \frac{\pi_\theta(y^+|x)}{\pi_{\text{ref}}(y^+|x)} - \log \frac{\pi_\theta(y^-|x)}{\pi_{\text{ref}}(y^-|x)}))] \tag{2}$$

where $\sigma$ is the sigmoid function, and $\mathcal{D}$ is the preference dataset. This objective encourages the model to assign higher probabilities to preferred completions.

**From pairwise to listwise preference**, LiPO (Liu et al., 2024d) extends DPO to handle ranked lists of responses $Y = \{y_1, ..., y_n\}$. It employs the pairwise logistic ranking loss (Burges et al., 2005) for sequence optimization. Specifically, each response is assigned a predicted score, defined as:

$$\{R_1, ..., R_n\} = \left\{ \log \frac{\pi_\theta(y_1|x)}{\pi_{\text{ref}}(y_1|x)}, ..., \log \frac{\pi_\theta(y_n|x)}{\pi_{\text{ref}}(y_n|x)} \right\} \tag{3}$$

To simplify notation, we use $R_*$ to represent these scores.

Additionally, each response is associated with a ranking level $\psi = \{\psi_1, ..., \psi_n\}$, which determines the sample's role in training: higher-ranked responses serve as positive samples, while lower-ranked ones are negative. The listwise ranking objective, in both its basic form and advanced variant (LiPO-$\lambda$), is defined as:

$$\mathcal{L}_{\text{LiPO}}(\theta) = -\mathbb{E}_{(x, Y, \psi) \sim \mathcal{D}} \left[ \sum_{\psi_i > \psi_j} \Delta_{i,j} \log \sigma\left(\beta(R_i - R_j)\right) \right] \tag{4}$$

In the basic version of LiPO, $\Delta_{i,j} = 1$ for all $i$ and $j$. In the advanced variant, $\Delta_{i,j}$, the Lambda weight, is used for more sophisticated preference pair weighting based on ranking levels.

Both methods enable efficient preference-based fine-tuning. LiPO offers more nuanced optimization by considering the relative rankings of multiple completions. These approaches align language models with human preferences without needing explicit reward modeling or reinforcement learning techniques.

## 3 PERPO: PERCEPTUAL PREFERENCE OPTIMIZATION

Motivated by the contrast between MLLMs' prowess in generative tasks (Yang et al., 2023b; Wen et al., 2024) and their struggles in visual discrimination (Li et al., 2024b; Qu et al., 2024), we aim to bridge this gap. We posit that this issue primarily stems from a lack of explicit perception alignment. Therefore, we employ preference optimization to simulate the human innate, coarse-to-fine visual perception process (Hegdé, 2008). As we will detail, we utilize the negative value of the model's prediction error relative to the visual ground truth as a reward signal. By maximizing the exploitation of this reward, we can effectively activate the model's inherent visual discrimination capability.

**A simple reward aligns well with visual discrimination.** The success of empirical risk minimization (ERM) (Pérez-Cruz et al., 2003; Golubev, 2004) in perceptual tasks (Zhang et al., 2018)

suggests the deterministic nature of ground truths in visual discrimination tasks. Practically, when a visual model is applied to well-defined discrimination tasks, generalization is often well-guaranteed. This indicates that the discrepancy between model predictions and ground truths can serve as a highly accurate and validated reward in visual discrimination tasks.

To substantiate this, Figure 1b visualizes the effects of Best-of-N (Charniak & Johnson, 2005; Nakano et al., 2021), SFT, DPO (Rafailov et al., 2024), and PerPO with $N$ samples, leveraging the model's object grounding performance on RefCOCOg (Mao et al., 2016). Among them, Best-of-N selects the answer with the highest reward, SFT uses the ground truth, DPO chooses the pair of answers with the largest reward discrepancy, and PerPO incorporates all answers. Notably, Best-of-N performance grows logarithmically with $N$, achieving 50% improvement at $N = 20$, demonstrating consistency between discriminative reward and model performance. In addition, DPO, trained on largest-margin pairs, surpasses SFT at $N = 8$, indicating the reward's efficacy in sample selection.

**Listwise rewarded samples boost visual preference optimization.** Methods like PPO (Schulman et al., 2017; Ouyang et al., 2022) and LiPO (Liu et al., 2024d) highlight the importance of diverse preference sample sequences in RL optimization. Generally, a sufficiently varied and systematically ordered set of negative samples helps the model rectify deficiencies incrementally and learn true preferences from rankings. Discriminative rewards, which require no human annotation, scale efficiently and enhance the impact of diverse negative samples for MLLMs. This is corroborated by Figure 1b, where PerPO's performance improves with increasing $N$. Table 4 further compares PerPO and DPO performance as $N$ increases, validating the superiority of listwise over pairwise negative sample optimization.

Meanwhile, recent studies show that human alignment in MLLMs doesn't effectively extend to visual conditions (Wang et al., 2024a), suggesting a form of image-unconditional reward hacking (Skalse et al., 2022). Our comparative analysis of DPO and PerPO, with and without image input (Figure 1c), reveals that PerPO exhibits superior gains with visual information. This indicates PerPO's optimization is more dependent on visual conditions. We attribute this robustness to the precision of discriminative reward and the strength of listwise optimization. For MLLMs, this implies that visual input engagement is crucial for accurate pattern identification.

**Your reward is secretly the perfect margin.** Often, rewards lack absolute values or have ambiguous magnitudes. Previous methods have addressed this by manually adding margins (Meng et al., 2024) or constructing imbalanced rankings based on permutations (Song et al., 2024) for balanced sorting. The success of these approaches fundamentally stems from the non-uniform objectives leading to smoother optimization spaces (Burges et al., 2006), although these spaces may not necessarily align with the preference space.

However, as mentioned earlier, the deterministic nature of discriminative rewards — specifically, the well-defined output space — ensures that we can guide an optimization space perfectly isomorphic to the discrimination space. Concretely, we use the absolute value of the reward itself as the weight for the sequence. Formally, we define $\left\{ \hat{R}_1, ..., \hat{R}_n \right\} = \{ f(x, y_1), ..., f(x, y_n) \}$ to denote the set of discriminative reward scores, where $\hat{R}_i$ is derived by evaluating the discrepancy (denoted by $f$) between sequence samples $Y$ and ground truth $x$. Based on them, we define the reward weight $w_{ij}$ for any pair of responses $(x, y_i, y_j)$ as:

$$w_{ij} = \frac{\left( \hat{R}_i - \hat{R}_j \right)^\gamma}{\sum_{\hat{R}_i > \hat{R}_j} \left( \hat{R}_i - \hat{R}_j \right)^\gamma} \tag{5}$$

where $\gamma$ is a scale factor. Notably, a norm design mitigates numerical impacts from varied discriminative rewards, enhancing model training robustness.

**The PerPO objective.** PerPO maximizes the ranking objective using discriminative reward scores to accurately measure response rankings. Leveraging these deterministic scores as the personalization reward weight for listwise preference amplifies the differences between distinct responses. Ultimately, the ranking optimization objective of our PerPO is defined as:

$$\mathcal{L}_{\text{PerPO}}(\theta) = -\mathbb{E}_{(x,Y)\sim\mathcal{D}}\Big[\sum_{\hat{R}_i > \hat{R}_j} w_{ij} \log \sigma(\beta(R_i - R_j))\Big] \qquad (6)$$

Overall, PerPO's listwise optimization intensifies penalties on negative samples, mitigating image-unconditional reward hacking, while refining performance through adaptive pairwise optimization based on discriminative rewards.

**Theoretically, PerPO is a listwise ERM.** A natural question is: *why don't we directly optimize discriminative rewards?* In other words, why not perform empirical risk minimization directly on MLLM? Interestingly, when we adjust the order of the discriminative reward margin and preference optimization objective in Eq 6, we have

$$\mathcal{L}_{\text{PerPO}}(\theta) = -\mathbb{E}_{(x,Y)\sim\mathcal{D}}\left[\sum_{\hat{R}_i > \hat{R}_j} \log \sigma(\beta(R_i - R_j)) \cdot \frac{\left(\hat{R}_i - \hat{R}_j\right)^{\gamma}}{\sum_{\hat{R}_i > \hat{R}_j}\left(\hat{R}_i - \hat{R}_j\right)^{\gamma}}\right] \qquad (7)$$

We can consider a simplified scenario where $\gamma$ equals 1 and $\sum_{\hat{R}_i > \hat{R}_j}\left(\hat{R}_i - \hat{R}_j\right)^{\gamma}$ is treated as a constant. In this case, Eq 7 expresses that for each $\hat{R}_i$, all $\hat{R}_m$ smaller than it form a coefficient in the preference optimization objective, while all $\hat{R}_n$ larger than it construct an opposite coefficient in this objective. Formally, this can be expressed as:

$$\mathcal{L}_{\text{PerPO}}(\theta) = -\mathbb{E}_{(x,Y)\sim\mathcal{D}}\left[\sum_{\hat{R}_i}\left(\sum_{\hat{R}_i > \hat{R}_m} \log \sigma(\beta(R_i - R_m)) - \sum_{\hat{R}_i < \hat{R}_n} \log \sigma(\beta(R_n - R_i))\right) \cdot \hat{R}_i\right]$$
$$(8)$$

we can observe that PerPO essentially implements a form of *listwise empirical risk minimization*. Each sample is assigned a dynamic weight, derived from the discriminative reward relationships between that sample and others. This weight is computed as the sum of preference optimization objectives based on the model's implicit reward $R$. This demonstrates **a coordination between discriminative rewards and the MLLM's inherent rewards**, theoretically proving PerPO's capability to model both visual discrimination and language generation abilities concurrently.

## 4 EXPERIMENTS

### 4.1 IMPLEMENTAL DETAILS

**Data construction.** We construct listwise preference data for two visual discriminative tasks: object grounding and dense OCR. Discriminative rewards are calculated using Intersection over Union (IoU) for object grounding and edit distance for dense OCR. For object grounding, we derive the corpus from RefCOCO (Yu et al., 2016), RefCOCO+ (Yu et al., 2016), and RefCOCOg (Mao et al., 2016). We sample an equal amount of data from each dataset and perform 20 samplings per instruction using the model at a temperature of 0.5. The resulting preference data are then filtered based on the data margin, defined as the difference between the maximum and minimum discriminative rewards within a list of responses. By setting the margin to 0.8, we retain 3,000 high-quality samples. For dense OCR, we use page-level OCR data from Fox (Liu et al., 2024a), employing edit distance instead of IoU for rewarding. Setting the margin to 0.04 yields a dataset of 1,800 samples.

**Models and training settings.** We adopt LLaVA-v1.5-7B (Liu et al., 2023a) as the base model, integrating CLIP-ViT-L-336px (Radford et al., 2021) and Vicuna-7B-v1.5 (Chiang et al., 2023; Liu et al., 2023b). All experiments are conducted using DeepSpeed ZeRO stage-3, applying LoRA (Hu et al., 2022) for fine-tuning. The training setup includes a batch size of 8 and a learning rate of 5e-6 with the AdamW optimizer. Training is completed on 8 GPUs in approximately 1.5 hours. To further validate our approach, we utilize LLaVA-Next-7B (Liu et al., 2024b) for both object grounding and dense OCR tasks. This model's sliced image processing capability enhances visual

Table 1: Performance comparison of SFT, DPO, and PerPO in object grounding and image understanding. **Bolding** indicates optimal performance, underlining indicates sub-optimal performance.

| Methods | RefCOCO | | | RefCOCO+ | | | RefCOCOg | | LLaVA$^W$ | MMHalBench | | POPE |
|---|---|---|---|---|---|---|---|---|---|---|---|---|
| | val | testA | testB | val | testA | testB | val | test | | Score ↑ | HalRate ↓ | |
| LLaVA-v1.5-7B | 50.0 | 59.9 | 43.3 | 45.8 | 55.2 | 34.6 | 49.4 | 49.3 | 61.8 | 2.11 | **0.54** | 86.1 |
| + SFT | 59.4 | 66.6 | 49.2 | 52.0 | 61.1 | 40.2 | 54.9 | 54.7 | 62.0 | 2.16 | 0.61 | 86.1 |
| + DPO | 60.6 | 67.8 | 50.5 | 53.3 | 62.1 | 41.4 | 55.9 | 55.1 | 61.3 | 2.08 | 0.62 | 86.3 |
| + PerPO | **63.8** | **70.6** | **54.4** | **57.3** | **65.9** | **46.9** | **60.0** | **59.6** | **64.0** | **2.26** | 0.57 | **86.5** |
| LLaVA-NEXT-7B | 84.9 | 90.5 | 77.3 | 77.6 | 86.8 | 67.0 | 80.7 | 80.3 | 72.7 | 2.79 | 0.48 | 87.5 |
| + SFT | 84.6 | 90.3 | 77.1 | 77.5 | 86.5 | 67.4 | 81.3 | 80.2 | 75.0 | 2.57 | 0.48 | **87.6** |
| + DPO | 85.5 | 90.8 | 78.8 | **78.1** | 86.9 | 68.0 | 81.0 | 81.1 | 77.6 | 2.69 | 0.49 | 87.5 |
| + PerPO | **86.7** | **91.3** | **81.0** | 69.4 | **87.3** | **70.1** | **82.4** | **82.4** | **81.2** | **2.81** | **0.46** | **87.6** |

Table 2: Performance comparison of SFT, DPO, and PerPO in dense OCR and image understanding. **Bolding** indicates optimal performance, underlining indicates sub-optimal performance.

| Methods | Edit Dist ↓ | F1 ↑ | Prec↑ | Rec ↑ | BLEU ↑ | METEOR ↑ | LLaVA$^W$ | MMHalBench | | POPE |
|---|---|---|---|---|---|---|---|---|---|---|
| | | | | | | | | Score ↑ | HalRate ↓ | |
| LLaVA-Next-25k-7B | 0.67 | 0.47 | 0.71 | 0.37 | 0.16 | 0.28 | **68.9** | 2.79 | 0.42 | 89.0 |
| + SFT | 0.66 | 0.47 | 0.72 | 0.38 | 0.17 | 0.29 | 67.8 | 2.85 | 0.42 | 89.0 |
| + DPO | 0.61 | 0.51 | **0.73** | 0.41 | 0.20 | 0.32 | 68.3 | **2.95** | 0.40 | 89.0 |
| + PerPO | **0.58** | **0.54** | **0.73** | **0.44** | **0.23** | **0.36** | 68.4 | 2.92 | **0.39** | 89.0 |
| LLaVA-Next-50k-7B | 0.64 | 0.51 | 0.74 | 0.41 | 0.18 | 0.31 | 70.2 | 2.97 | 0.36 | 89.6 |
| + SFT | 0.62 | 0.52 | 0.74 | 0.42 | 0.20 | 0.32 | 69.8 | **3.15** | **0.34** | 89.9 |
| + DPO | 0.60 | 0.54 | **0.75** | 0.43 | 0.21 | 0.33 | 69.2 | 3.10 | 0.36 | **90.0** |
| + PerPO | **0.56** | **0.56** | **0.75** | **0.46** | **0.24** | **0.36** | **71.5** | 3.00 | 0.36 | **90.0** |

understanding. However, it demonstrates limited efficacy in the dense OCR task, likely due to a lack of sufficient training data. To address this, we construct page OCR datasets of varying sizes (25k, 50k), combining them with the original 780k instruction tuning data to train LLaVA-Next-*k-7B. Unlike previous models, this version employs SigLIP-400M (Zhai et al., 2023) as the visual encoder and Qwen2-7B (Yang et al., 2024) as the language model.

**Evaluation benchmarks.** We conduct a comprehensive assessment of PerPO across various multimodal benchmarks. Using LLaVA$^W$ (Liu et al., 2023a), we evaluate the general capabilities of multimodal models. To assess perceptual robustness, we employ hallucination metrics from MMHalBench (Sun et al., 2023) and POPE (Li et al., 2023). For object grounding, we utilize the RefCOCO, RefCOCO+, and RefCOCOg datasets, with AP@50 as the evaluation metric. In the dense OCR scenario, we use Fox's proprietary dataset, measuring performance with Edit Distance, F1-score, Precision, Recall, BLEU (Papineni et al., 2002), and METEOR (Satanjeev, 2005). Meanwhile, Appendix A.2 provides additional metrics for evaluating the model's performance in general visual tasks. This comprehensive evaluation provides valuable insights into PerPO's capacity in addressing multimodal challenges.

## 4.2 PERFORMANCE COMPARISON

**Superior performance of PerPO across various visual discriminative tasks.** To demonstrate PerPO's effectiveness, we evaluate SFT, DPO and our PerPO on different model baselines across various downstream tasks. As shown in Table 1, PerPO consistently outperforms SFT and DPO across benchmarks, revealing a superiority of listwise preference optimization to pointwise (SFT) and pairwise (DPO). On LLaVA-v1.5-7B, PerPO significantly boosts the object grounding capacity, with relative gains of 3.42%, 8.18%, and 5.58% on RefCOCO, RefCOCO+, and RefCOCOg, re-

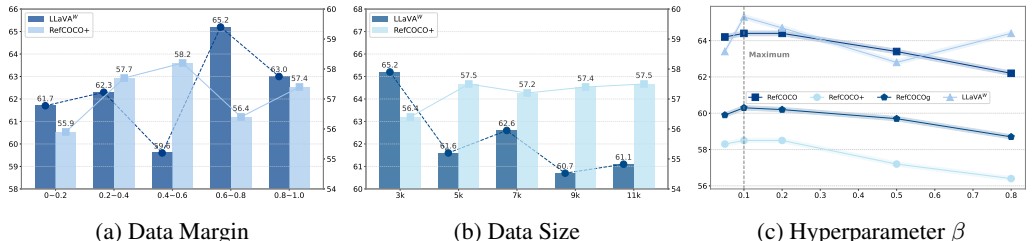

(a) Data Margin  (b) Data Size  (c) Hyperparameter $\beta$

Figure 2: Analysis of training data quality, quantity, and hyperparameter $\beta$ (a) Performance across different data margins. (b) Performance across different data sizes. (c) Performance across different $\beta$ values in the loss function.

spectively. On a stronger baseline LLaVA-NEXT-7B, PerPO also delivers consistent improvements, demonstrating its cross-model generalizability. PerPO similarly demonstrates its superiority in the highly applicable dense OCR scenario. Table 2 illustrates this by showing significant reductions in edit distance on two baselines (13.4% in LLaVA-Next-25k-7B and 14.3% in LLaVA-Next-50k-7B, respectively). This highlights, first, PerPO's cross-task generalizability, and second, its higher data utilization efficiency compared to SFT and DPO.

**PerPO also improves general image understanding.** As demonstrated in Table 1 and Table 2, PerPO exhibits substantial improvements in general image understanding (LLaVA$^W$) and image hallucination mitigation (MMHalBench and POPE). This indicates that despite PerPO's singular focus on aligning perceptual processes, it effectively generalizes to broader image comprehension domains, and in fact, deepens image cognition.

### 4.3 ABLATION STUDY

**Training data statistical analysis.** Training data plays a crucial role in preference optimization. We conduct a comprehensive statistical analysis, focusing on data quality and quantity. Quality is assessed by the margin, defined as the difference between the highest and lowest discriminative scores within a list. As shown in Figure 2a, the experimental results are influenced by the margin. A balanced performance for both LLaVA$^W$ and RefCOCO+ is achieved with the margin of 0.8 to 1.0. Figure 2b indicates that RefCOCO+ improves with larger data size, while LLaVA$^W$ declines. Optimal performance occurs at 3k samples.

**Hyperparameter $\beta$ in PerPO loss.** DPO loss includes a hyperparameter $\beta$, which controls the model's sensitivity to differences between candidate responses. A higher $\beta$ increases the model's focus on subtle distinctions in outputs, while a lower $\beta$ allows for greater tolerance of minor deviations. During training, $\beta$ also affects the model's rate of assimilating human preferences, with an optimal value ensuring stable learning progression. This parameter, also applied in our PerPO method, underwent several experimental iterations. As shown in Figure 2c, the best performance was achieved with $\beta$ set to 0.1.

Table 3: Analysis of LoRA training strategy.

| $r$ | $\alpha$ | Ref | Ref+ | Refg | LLaVA$^W$ | POPE |
|---|---|---|---|---|---|---|
| 64 | 128 | 62.9 | 57.0 | 59.5 | 62.2 | 86.4 |
| 128 | 256 | 63.4 | 57.2 | 59.7 | 62.8 | 86.4 |
| 256 | 512 | 63.7 | 57.6 | 60.0 | 64.1 | 86.5 |
| 512 | 1024 | 64.4 | 58.2 | 60.3 | **64.6** | **86.7** |
| 1024 | 2048 | **65.8** | **59.6** | **61.5** | 64.2 | 86.6 |

**LoRA training strategy.** The calibration of hyperparameters $r$ and $\alpha$ in LoRA training illustrates the balance between specialized learning and general competence in fine-tuning. Higher $r$ values enhance task-specific knowledge acquisition but carry the risk of catastrophic forgetting, while $\alpha$ controls the magnitude of weight updates. As demonstrated in Table 3, the horizontal and vertical axes represent the values of LLaVA$^W$ and RefCOCO, respectively. As $r$ increases, the model's performance shows an upward trend. Our experiments with PerPO, conducted at $r = 128$ and $\alpha = 256$, prioritize computational efficiency over maximizing performance, in order to reduce resource consumption. This approach underscores the trade-off between theoretical optimization and computational constraints in applied machine learning.

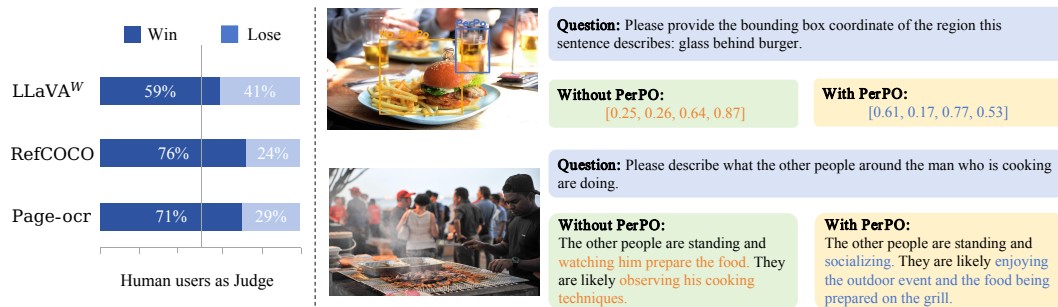

Figure 3: Relative performance (Left, Human users as judge) and comparative showcases (Right) with and without PerPO alignment across different tasks.

Table 4: Performance comparison of PerPO and DPO for different sample sizes $N$. **Bolding** indicates optimal performance, underlining indicates sub-optimal performance.

| $N$ | Methods | Ref+ | Refg | LLaVA$^W$ | POPE | Methods | Ref+ | Refg | LLaVA$^W$ | POPE |
|---|---|---|---|---|---|---|---|---|---|---|
| 2 | DPO | 50.9 | 54.0 | 60.1 | 86.2 | PerPO | 55.4 | 57.3 | **65.9** | 86.3 |
| 4 | DPO | 52.2 | 54.6 | 60.6 | **86.3** | PerPO | 56.2 | 58.6 | 61.2 | **86.5** |
| 8 | DPO | 52.6 | 55.2 | 62.4 | 86.2 | PerPO | 57.0 | 59.3 | 62.1 | 86.4 |
| 12 | DPO | 52.7 | 55.4 | **62.6** | 86.2 | PerPO | **57.4** | 59.4 | 63.1 | **86.5** |
| 20 | DPO | **52.9** | 55.4 | 61.2 | 86.2 | PerPO | **57.4** | 59.7 | 64.7 | **86.5** |

## 5 IN-DEPTH ANALYSIS

### 5.1 IMPACT OF DISCRIMINATIVE REWARD IN PERPO

**Discriminative reward aligns well with perception.** We conducted a comparative analysis of Best-of-N, SFT, DPO, and PerPO on object grounding task, using IoU as discriminative reward. To explore upper-bound performance, we calculated Best-of-N using test set ground truth, while other methods utilized the train set. Sampling was performed at temperature 0.5 from a moderately capable model. As shown in Figure 1a, Best-of-N's logarithmic performance trend with increasing samples validates the reward's effectiveness in aligning with perception performance in an oracle scenario. Meanwhile, the enhanced gains of DPO and PerPO at higher $N$ values confirm the accuracy of reward-based sample selection or ranking, highlighting the potential of reward-guided approaches for model improvement.

**Discriminative reward also aligns well with human.** To assess PerPO's user alignment, we employed both GPT-4o and human users to compare models before and after PerPO alignment from multiple perspectives. We uniformly sampled 500 questions from open-ended datasets like LLaVA$^W$, RefCOCO, and Page-ocr in Fox, and evaluated relative performance, considering response accuracy, instruction adherence, and hallucination reduction. A more detailed description of the evaluation can be found in Appendix A.3. Figure 3 (left) shows that the PerPO-aligned model achieved a higher win rate, with significant improvements in different datasets. Therefore, enhancing perception not only aligns better with human preferences but also boosts user experience due to stronger visual capabilities and more efficient optimization.

### 5.2 IMPACT OF LISTWISE PREFERENCE IN PERPO

**More negative supervisions help discrimination.** Figure 1b illustrates the asymptotic growth of DPO and PerPO under increased sampling, preliminarily validating the value of negative samples. We further conduct a comprehensive comparison between PerPO and DPO across multiple benchmarks including RefCOCO+, RefCOCOg, LLaVA$^W$, and POPE, examining performance disparities at varying sample sizes $2, 4, 8, 12, 20$. In Table 4, observations reveal that increased sampling consistently led to improved performance across diverse metrics. Moreover, PerPO demonstrated more

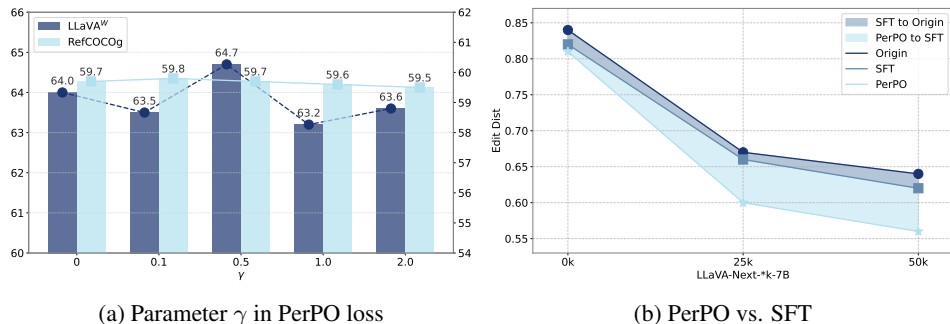

(a) Parameter $\gamma$ in PerPO loss            (b) PerPO vs. SFT

Figure 4: (a) Performance across different $\gamma$ values in PerPO loss. (b) Comparison of PerPO and SFT across different dense OCR levels. As the model capability increases and approaches saturation, PerPO can unleash the full potential of the model compared to SFT.

pronounced absolute performance and performance gains relative to DPO. This confirms the role of negative sample supervision in visual preference optimization. Notably, as sampling size $N$ increases, performance gains saturate, indicating a loss of negative sample diversity. Thus, mining more diverse negative samples is critical and will be pursued in future work.

**Listwise preference optimization helps prevent image-unconditional reward hacking.** As discussed in Section 3, we compared the preference optimization results of DPO and PerPO with and without image input on RefCOCOg and LLaVA$^W$. PerPO shows significant performance gains over DPO with image input, demonstrating that PerPO's optimization is more reliant on visual conditions, and hence helps prevent such reward hacking.

### 5.3 IMPACT OF DISCRIMINATIVE MARGIN.

**Reward itself serves as the perfect margin.** As shown in Eq 6, we introduce a coefficient $\gamma$ to finely modulate the influence of the differential discriminative rewards on the corresponding sample pairs. It can be seen that when $\gamma = 0$, PerPO simplifies to LiPO. When $\gamma \neq 0$, unlike LiPO balanced ranking, PerPO can emphasize inter-sample distinctions, facilitating more targeted optimization. Our ablation study on $\gamma$ parameter, presented in Figure 4a, shows that the model achieves optimal performance at $\gamma = 0.5$, highlighting the effectiveness of our personalized weighting strategy in improving model performance.

### 5.4 FURTHER ANALYSIS

**PerPO aims to unlock the model's full potential.** PerPO's effectiveness seems to depend on the capability level of the model. Comparing SFT and PerPO performance on models trained with varying amounts of OCR data (0k, 25k, 50k), we found that PerPO's advantage emerges only as the model's capabilities mature. Figure 4b shows that with weak or no dense OCR capabilities, PerPO and SFT perform similarly. However, as the model approaches capability saturation, the area of the light blue region increases significantly, indicating that PerPO outperforms SFT. To sum up, SFT is crucial for imparting basic capabilities, whereas PerPO is key to unlocking the model's full potential in later stages.

**Qualitative analysis.** To qualitatively analyze the effectiveness of PerPO, as shown in Figure 3 (right), we present two cases highlighting the differences before and after applying PerPO. The first case involves the object grounding task of locating a glass behind a hamburger. Initially, the model focuses on the hamburger, but after alignment, it correctly identifies the glass. The second case is to ask what the other people arounding the man cooking in the image are doing. Without PerPO, the model would mistakenly think they are watching the man prepare the food and observing his cooking techniques, while the model with PerPO would answer that the people around are socializing and they are enjoying outdoor event and the food being prepared on the grill. PerPO not only improves the accuracy of visual recognition tasks such as object detection, but also reduces hallucinations and enhances visual perception capabilities.

## 6    RELATED WORK

**Reinforcement Learning from Human Feedback (RLHF).** RLHF (Christiano et al., 2017; Stiennon et al., 2020) is a crucial technique for aligning Large Language Models (LLMs) with human preferences, comprising both reward model-based and model-free methods. In PPO (Schulman et al., 2017; Ouyang et al., 2022), an auxiliary reward model is cultivated first and then used to optimize the policy. Conversely, DPO (Rafailov et al., 2024) directly leverages preference data for policy optimization, offering a streamlined yet effective pathway for alignment. To mitigate overfitting, IPO (Azar et al., 2024) incorporates a regularization term. KTO (Ethayarajh et al., 2024) and DPOP (Pal et al., 2024) optimize the relative gain of outputs, bypassing the need for pairwise data. sDPO (Kim et al., 2024) uses multi-stage training for better alignment. ORPO (Hong et al.) and SimPO (Meng et al., 2024) adopt reference-free reward formulations to simplify alignment. Despite impressive results, these methods rely on labeled perference data, limiting their generalizability. In contrast, PerPO uses a discriminative reward mechanism, allowing data scaling without extra costs and enhancing model performance across diverse domains.

**Multimodal Large Language Models (MLLMs).** MLLMs (Liu et al., 2024c; Yu et al., 2023; Zhu et al., 2024; Dong et al., 2024; Ghosal et al., 2023; Lin et al., 2023) integrate various data modalities into a unified framework, enabling more sophisticated content understanding and generation. Vision-Language Models (VLMs) are a prominent example, aligning visual encoders with LLMs to connect different modal information. Recently, MLLMs have been evolving to enhance reliability and incorporate ethical considerations, aiming to align their outputs with human values (Amirloo et al., 2024; Yu et al., 2024a; Xu et al., 2024). LLaVA-RLHF (Sun et al., 2023) leverages supplementary factual information to enhance the reward model, mitigating vulnerabilities like reward hacking. HA-DPO (Zhao et al., 2023) reframes hallucination as a preference task, introducing an efficient pipeline for generating high-quality, consistent sample pairs. Additionally, mDPO (Wang et al., 2024a) balances language and image preferences, reducing the over-emphasis on textual inputs. Nevertheless, these models focus on reasoning and reducing hallucinations, they often struggle with discriminative tasks requiring minimal analysis and concise answers. PerPO, however, can enhance models' visual comprehension abilities through discriminative rewards.

**Generation and Discrimination.** AI's landscape is shaped by discriminative tasks, which classify and predict (Godbole & Sarawagi, 2004; Bhat et al., 2019; Zhu et al., 2021), and generative tasks, which create and innovate (Radford, 2018; Radford et al., 2019). Traditionally distinct, these tasks are now converging in the era of MLLMs. Hybrid applications, such as conversational agents (Brown, 2020; Nguyen, 2023; Wölfel et al., 2024) that understand and generate text or autonomous vehicles (Schwarting et al., 2018; Janai et al., 2020; Wang et al., 2021) that recognize objects and make decisions, exemplify this trend. Discriminative tasks are increasingly tackled through generative modeling, yielding impressive results in areas like mathematical reasoning (Cobbe et al., 2021; Shi et al., 2024) and multimodal inference (Zhao et al., 2024a; Wang et al., 2024b). However, current MLLM architectures face limitations in visual discrimination due to the absence of negative reinforcement. PerPO addresses this shortcoming by optimizing perceptual ordered preferences from discriminative rewards, effectively bridging the gap between MLLMs' generative prowess and their discriminative capabilities in visual tasks.

## 7    DISCUSSION

**Conclusion.** In this paper, we highlight the limitations of Multimodal Large Language Models (MLLMs) in visual discrimination tasks, such as object recognition and dense OCR. Therefore, we propose Perceptual Preference Optimization (PerPO), a novel framework that enhances the visual discrimination capabilities of MLLMs through discriminative rewarding. By constructing perceptual ordered preferences based on prediction deviations, the performance is effectively optimized without the need for extensive human annotations. The extensive experiments on widely-used benchmarks demonstrate that PerPO not only significantly improves the performance of MLLMs and the output robustness in visual tasks. The innovative method bridges the gap between generative and discriminative functionalities, paving the way for more comprehensive artificial intelligence systems that can excel in both creative generation and perceptual understanding.

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

Table 5: Performance comparison of SFT, DPO, and PerPO in object grounding and image understanding. **Bolding** indicates optimal performance, underlining indicates sub-optimal performance.

| Methods | RefCOCO | | | RefCOCO+ | | | RefCOCOg | | LLaVA$^W$ | MMHalBench | | POPE |
| | val | testA | testB | val | testA | testB | val | test | | Score ↑ | HalRate ↓ | |
|---|---|---|---|---|---|---|---|---|---|---|---|---|
| LLaVA-OneVision | 73.6 | 82.6 | 63.8 | 69.4 | 79.5 | 58.2 | 71.1 | 70.8 | 79.7 | 2.70 | 0.41 | 88.3 |
| + SFT | 74.7 | 83.7 | 65.4 | 70.3 | 80.8 | 59.1 | 72.1 | 71.7 | 77.9 | 2.73 | 0.40 | 88.1 |
| + DPO | 79.5 | 86.5 | 71.1 | 74.6 | 83.4 | 64.5 | 76.3 | 76.1 | 80.1 | 2.75 | 0.39 | 88.4 |
| + PerPO | **82.2** | **88.1** | **75.6** | **77.3** | **85.3** | **68.4** | **79.6** | **79.9** | **83.3** | **2.82** | **0.37** | **88.8** |

Table 6: Performance comparison of SFT, DPO, and PerPO on general visual benchmarks.

| Methods | MM-Vet | MM-Bench | MMMU | VQAv2 | LLaVA$^W$ |
|---|---|---|---|---|---|
| LLaVA-v1.5-7B | 32.9 | 62.3 | 35.7 | 78.5 | 61.8 |
| + SFT | 31.0 | 62.5 | 36.7 | 78.6 | 62.0 |
| + DPO | 31.2 | 62.3 | 36.0 | 78.4 | 61.3 |
| + PerPO | **33.3** | **62.8** | **37.0** | **78.8** | **64.0** |

## A A COMPREHENSIVE ASSESSMENT OF PERPO

### A.1 GENERALIZATION ASSESSMENT

**Performance on LLaVA-OneVision (Li et al., 2024a).** To assess PerPO's generalization capability, we performed comparative experiments on LLaVA-OneVision for object grounding. We initially constructed model-specific datasets by leveraging the diverse responses, retaining 3k listwise preference data, after filtering. Detailed results are shown in Table 5. It is evident that after perceptual alignment training, the model show improvements in both specific and general capabilities, significantly surpassing SFT and DPO. Extensive experimentation conclusively demonstrates PerPO's robust generalization capabilities.

### A.2 GENERAL VISUAL CAPACITY ASSESSMENT

Our method enhances model perception by employing discriminative rewards in specific tasks like object grounding and dense OCR. To thoroughly evaluate PerPO's capabilities on general visual tasks, we included diverse benchmarks in Table 6, such as **MM-Vet (Yu et al., 2024b), MM-Bench (Liu et al., 2024e), MMMU (Yue et al., 2024), VQAv2 (Goyal et al., 2017), and LLaVA$^W$ (Liu et al., 2023a)**. The results clearly demonstrate a significant advantage over SFT and DPO, confirming PerPO's superior efficacy.

**MM-Vet** stands as a preeminent multimodal evaluation metric, critically assessing models across six dimensions: recognition, OCR, knowledge, language generation, spatial reasoning, and mathematical computation. Detailed evaluation results within MM-Vet are presented in Table 7. Obviously, our method excels across multiple tasks, indirectly suggesting an enhancement in the model's perceptual capabilities.

**MM-Bench** is designed to systematically evaluate multimodal models on a range of vision-language tasks with emphasis on robustness, reasoning, and generalization. It often focuses on benchmarks that highlight deficiencies in current vision-language systems. Detailed evaluation criteria and associated tasks span domains like captioning, VQA, and multimodal reasoning.

**MMMU** stands for multimodal multitask understanding, encompassing datasets and benchmarks tailored to models capable of performing multiple tasks. It is a concept designed to focus on advanced perception and reasoning with domain-specific knowledge, emphasizing flexibility and comprehension across various visual and linguistic scenarios.

Table 7: Performance comparison of SFT, DPO, and PerPO on MM-Vet.

| Methods | Rec | Ocr | Know | Gen | Spat | Math | Overall |
|---|---|---|---|---|---|---|---|
| LLaVA-v1.5-7B | 44.9 | 26.7 | **22.9** | 21.5 | 25.6 | 7.7 | 32.9 |
| + SFT | 43.8 | 25.6 | 16.7 | 20.6 | 24.9 | 7.7 | 31.0 |
| + DPO | 43.5 | 24.6 | 19.5 | 22.5 | 24.5 | 7.7 | 31.2 |
| + PerPO | **45.1** | **29.3** | 19.5 | **23.0** | **26.8** | **12.7** | **33.3** |

Table 8: The evaluation of GPT-4o and Human users.

| | LLaVA$^W$ | RefCOCO | Page-ocr |
|---|---|---|---|
| Win rate as judged by GPT-4o | 56% | 72% | 71% |
| Win rate as judged by Human users | 59% | 76% | 71% |

**VQAv2** is a dataset for visual question answering, addressing issues like biases in earlier datasets. It contains pairs of images and questions with answers verified by human annotators, ensuring higher reliability and reducing the tendency of models to exploit statistical patterns in the dataset.

**LLaVA$^W$** evaluates multimodal large language models on real-world, unstructured inputs like everyday photos and screenshots. It focuses on tasks such as visual question answering, reasoning, and conversational understanding, using human and AI feedback to assess accuracy and relevance. This benchmark emphasizes practical robustness in diverse, open-world applications.

## A.3 GPT-4O AND HUMAN USERS ASSESSMENT

We conducted a comparative analysis of models before and after PerPO alignment, utilizing assessments from GPT-4o and human users across three dimensions: response accuracy (RA), instruction adherence (IA), and hallucination reduction (HaR). The test dataset comprises 500 samples sourced from multiple public datasets. Ultimately, we derived the win rates for PerPO across individual datasets in Table 8. The results indicate that the evaluations of GPT-4o and humans yield relatively consistent outcomes.

**GPT-4o prompt template.** The prompt used to compare the responses before and after applying PerPO is illustrated in Figure 5.

**Human users.** We invited 20 experts and scholars specializing in computer vision, natural language processing, and human-computer interaction to provide independent assessments. For each question, we calculated the average scores in terms of response accuracy, instruction adherence, and

---

**GPT-4o for Assessment**

As a professional evaluator of computer vision and natural language processing data, you will be presented with an image, a question, and two corresponding answers. Please rate each response on the following three aspects, using a scale from 5 to 1 (5 indicating highly satisfactory, 4 satisfactory, 3 uncertain, 2 somewhat unsatisfactory, and 1 completely unsatisfactory).

1. Response Accuracy: The content of the response is correct based on the provided image and question, ensuring image-text consistency and coherence.
2. Instruction Adherence: The response strictly follows user instructions, carefully addresses each question posed by the user, and outputs in the format requested.
3. Hallucination Reduction: The response content is credible and authentic, with minimal provision of false information.

Additionally, based on the above ratings, you need to select the response you consider superior. Output 1 if the first response is better, 2 if the second response is better, or 0 if both responses are equally good. Please ensure that your final output adheres to the following format. Maintain output conformity and don't provide extra output.

OUTPUT:
Response1: [Response Accuracy: #<A score>#, Instruction Adherence: #<A score>#, Hallucination Reduction: #<A score>#]
Response2: [Response Accuracy: #<A score>#, Instruction Adherence: #<A score>#, Hallucination Reduction: #<A score>#]
The selected response: #<selected response>#

Figure 5: The prompt for comparing the responses before and after applying PerPO.

hallucination reduction. The winning response was determined based on the magnitude of these average scores. Finally, we aggregated evaluations from 20 expert assessors to determine PerPO's overall win rate.

## B LIMITATION AND FUTURE WORK

While PerPO has significantly advanced the visual discrimination capabilities of MLLMs, it still has some limitations. The better effectiveness may depend on the support of specific datasets, limiting the generalizability of performance. Additionally, although it reduces reliance on human annotations, more complex tasks may still require human annotations for more precise feedback. In the future, we will further explore the implications of PerPO across various applications to fully realize the potential of MLLMs in diverse domains. Moreover, the combination with other advanced innovations will be developed for better overall model performance.

