# OpenReview forum: "PerPO: Perceptual Preference Optimization via Discriminative Rewarding"
_ICLR.cc/2025/Conference — Submitted to ICLR 2025_

### Official Review · Reviewer_oqhZ · 2024-11-02

**Soundness:** 3
**Presentation:** 2
**Contribution:** 2
**Rating:** 5
**Confidence:** 3

**Summary:**

This paper introduces Perceptual Preference Optimization (PerPO), a method designed to enhance the visual discrimination abilities of multimodal large language models (MLLMs). By using discriminative rewards and listwise preference optimization, PerPO aligns MLLMs with human perception, addressing issues like image-unconditional reward hacking. It improves MLLM performance in visual tasks, such as object grounding and OCR, while maintaining generative capabilities and demonstrating generalizability across models and tasks.

**Strengths:**

1. This paper addresses a unique and intriguing research question by exploring the discriminative abilities of multimodal large language models (MLLMs), a direction not commonly emphasized in current studies.
2. Experiments demonstrate PerPO's superiority over baseline techniques like Direct Preference Optimization (DPO) across multiple benchmarks and visual tasks, highlighting its broad applicability and effectiveness.

**Weaknesses:**

1. The paper argues that the primary limitation in the visual discrimination capabilities of MLLMs is due to a lack of explicit perceptual alignment, but it does not provide sufficient evidence to support this claim. There is no experimental evidence that this issue arises from the alignment stage rather than other factors such as instruction tuning or an insufficient pre-training dataset. I recommend that the authors present more compelling evidence to confirm the assertion that the main challenge in the "limitations in MLLMs' visual discrimination" is mainly related to "perceptual alignment."
2. The current method may encounter limitations when applied to complex visual tasks that require significant contextual understanding. While these challenges are briefly mentioned, they are not fully investigated in the study.

**Questions:**

1. In Table 1, the performance of the LLaVA-Next model with PerPO shows a noticeable decrease. Could the authors explain the reasons behind this performance drop?
2. Could the authors provide results on the effect of scaling up the model on the discriminative reward?

---

> ### Author Response · Authors · 2024-11-22
> **Response to Reviewer oqhZ**
>
> We sincerely appreciate your careful and constructive reviews. We have responded to each of your comments and questions as follows:
>
> ***W1: The paper argues that the primary limitation in the visual discrimination capabilities of MLLMs is due to a lack of explicit perceptual alignment, but it does not provide sufficient evidence to support this claim. There is no experimental evidence that this issue arises from the alignment stage rather than other factors such as instruction tuning or an insufficient pre-training dataset. I recommend that the authors present more compelling evidence to confirm the assertion that the main challenge in the "limitations in MLLMs' visual discrimination" is mainly related to "perceptual alignment."***
>
> Thank you for pointing out this issue. Indeed, we would like to emphasize the following two points:
>
> 1. We acknowledge the importance of pre-training and instruction fine-tuning for visual discrimination. However, we **emphasize that PerPO aims to push the boundaries and further unlock perceptual potential**. Despite extensive training, many models, like GPT-4o, struggle with visual tasks such as object grounding and boundary drift, highlighting the limitations of current models. In deterministic tasks like mathematical reasoning, language models leverage strategies like *diverse negative supervision* [1] or *fine-grained process rewarding* [2] through RLHF to exploit intrinsic capabilities. This analogy supports our approach. Moreover, as shown in Figure 1b, the Best-of-N results reveal the model’s greater potential, validating our method. Therefore, we propose PerPO to activate the visual perceptual potential acquired during pre-training and instruction tuning, addressing gaps left by existing methods.
>
> 2. Compared to pre-training and instruction fine-tuning, **PerPO demonstrates significant advantages in data <u>construction costs</u> and data <u>utilization efficiency</u>.** First, while pre-training and instruction fine-tuning enhance model perception, they demand large, precise datasets. In contrast, PerPO employs a **scalable** approach for rapid, **cost-free** data acquisition based on *temperature sampling* and *discriminative rewarding*. Second, as shown in the table below, doubling dense OCR data from 25k to 50k yields a slight edit distance improvement from 0.67 to 0.64. However, PerPO, using only 1.8k preference data for LLaVA-Next-25k-7B model, significantly reduces the edit distance to 0.58. This demonstrates PerPO’s ability to **substantially boost performance** while **minimizing training costs**.
>
>     | Methods | Edit Dist ↓ | F1 ↑  | Prec ↑ | Rec ↑ | BLEU ↑ | METEOR↑|
>     |-----------------------|-------------|-------|--------|-------|--------|----------|
>     | LLaVA-Next-25k-7B     | 0.67 | 0.47  | 0.71   | 0.37  | 0.16   | 0.28 |
>     | **+ PerPO**  | **0.58**    | **0.54**| **0.73**| **0.44**| **0.23**| **0.36**|
>     | LLaVA-Next-50k-7B     | 0.64 | 0.51  | 0.74   | 0.41  | 0.18   | 0.31 |
>     | **+ PerPO**  | **0.56**    | **0.56**| **0.75**| **0.46**| **0.24**| **0.36**|
>
> **References**
>
> [1] Shao et al. Deepseekmath: Pushing the limits of mathematical reasoning in open language models.
> [2] Setlur et al. Rewarding Progress: Scaling Automated Process Verifiers for LLM Reasoning.

---

> > ### Author Response · Authors · 2024-11-22
> > **Response to Reviewer oqhZ**
> >
> > ***W2: The current method may encounter limitations when applied to complex visual tasks that require significant contextual understanding. While these challenges are briefly mentioned, they are not fully investigated in the study.***
> >
> > We appreciate your insights. Firstly, we highlight our model’s enhanced performance using PerPO on complex visual tasks. We also acknowledge the challenge of constructing listwise preference data for complex discriminative tasks and propose a solution in the second point. We hope our response meets your expectations.
> >
> > 1. To validate PerPO's effectiveness in complex visual tasks, we present results from **MM-Vet, MM-Bench, MMMU, VQAv2, and LLaVA^(W)** benchmarks in the first table below. Among these, MM-Vet serves as a particularly comprehensive evaluation, assessing the model’s performance across six visual subtasks, including **recognition, OCR, knowledge, language generation, spatial awareness, and mathematical reasoning**. The second table provides detailed subtask results for MM-Vet. Notably, our model, trained exclusively on task-specific data, exhibits improved performance in more complex visual scenarios, indirectly validating that this approach enhances model's capacity to perceive and comprehend images.
> >
> >     | Methods | MM-Vet |MM-Bench| MMMU  | VQAv2 | LLaVA^(W) |
> >     |---|--|--|--|---|---|
> >     | LLaVA-v1.5-7B |  32.9 | 62.3 | 35.7 | 78.5  | 61.8 |
> >     | + SFT | 31.0 | 62.5 | 36.7 |  78.6  | 62.0 |
> >     | + DPO | 31.2 | 62.3 | 36.0 | 78.4  | 61.3 |
> >     | + PerPO | **33.3** | **62.8** | **37.0** | **78.8**  | **64.0** |
> >
> >     | Methods | Rec | Ocr| Know | Gen | Spat | Math |   Overall |
> >     |---|--|--|--|---|---|---|---|
> >     | LLaVA-v1.5-7B | 44.9  | 26.7 | **22.9**  | 21.5 | 25.6  |  7.7 | 32.9 |
> >     | + SFT | 43.8 | 25.6 | 16.7 | 20.6  |  24.9 |  7.7 | 31.0 |
> >     | + DPO | 43.5 | 24.6 | 19.5| 22.5 | 24.5 | 7.7 | 31.2 |
> >     | + PerPO | **45.1** | **29.3** | 19.5 | **23.0** | **26.8** | **12.7**   | **33.3** |
> >
> > 2. To align with a broader range of visual scenarios in the future, we intend to **integrate multiple expert models for data construction and incorporate diverse metrics as our discriminative rewards**. Specifically, we can emulate approaches such as PaLI-X [1], Qwen-VL [2], InternVL [3], and LLaMA3.2-Vision [4] by using expert-level OCR models and open-vocabulary detectors to first generate detailed and fine-grained pseudo-labels. Subsequently, we can develop a composite discriminative reward centered around algorithms such as IoU, edit distance, and object matching to evaluate the consistency of model outputs against expert pseudo-labels. Finally, leveraging this reward, we can sample diverse preference data to achieve perceptual alignment in complex visual tasks.
> >
> > **References**
> > [1] Chen et al., PaLi-X: On scaling up a multilingual vision and language model.
> > [2] Bai et al., Qwen-vl: A frontier large vision-language model with versatile abilities.
> > [3] Chen et al., Internvl: Scaling up vision foundation models and aligning for generic visual-linguistic tasks.
> > [4] Meta AI, Llama 3.2: Revolutionizing edge AI and vision with open, customizable models.

---

> ### Author Response · Authors · 2024-11-22
> **Response to Reviewer oqhZ**
>
> ***Q1: In Table 1, the performance of the LLaVA-Next model with PerPO shows a noticeable decrease. Could the authors explain the reasons behind this performance drop?***
>
> We are uncertain about the performance drop you mentioned: whether it refers to the decreased improvement of LLaVA-Next over LLaVA-v1.5 under the PerPO setting or the decline in LLaVA-Next’s metrics on the RefCOCO+ validation set. We address both phenomena separately in the following text.
>
> - LLaVA-Next, enhanced with augmented training data and a refined image processing paradigm, naturally exhibits commendable object grounding capabilities. This, to some extent, constrains the potential improvements achievable through post-training. In contrast, LLaVA-v1.5, owing to its comparatively weaker inherent capacities, offers ample room for enhancement via PerPO. In summary, **the effectiveness of PerPO training is closely tied to the model’s innate capabilities**. Nonetheless, PerPO consistently delivers significant improvements over SFT and DPO across all configurations.
>
> - Despite our method’s general superiority, we noted a performance dip on the RefCOCO+ validation set. Analysis revealed two likely causes:
>   1. **Underrepresentation in Preference Data**: RefCOCO+ constitutes less than 25% of our constructed preference dataset, significantly lower than RefCOCO and RefCOCOg. This disparity may lead to suppression of its unique representation.
>   2. **Distinct Prompt Distribution**: RefCOCO+ exhibits a markedly different prompt distribution compared to RefCOCO and RefCOCOg. Specifically, RefCOCO and RefCOCOg focus more on queries incorporating spatial relationships (e.g., indicating an object’s location on the left side of an image). In contrast, RefCOCO+ emphasizes queries about absolute object positions.
>
>   The combination of this training discrepancy and the smaller representation results in a degree of ***distribution shift*** during evaluation. Nevertheless, the model’s consistent performance on RefCOCO+'s testA and testB sets demonstrates its inherent robustness.
>
> ***Q2: Could the authors provide results on the effect of scaling up the model on the discriminative reward?***
>
> According to your suggestions, we have supplemented the experimental results on the LLaVA-Next-13B model. The comparative results are shown in the table below. As can be seen from the table, **the model continues to achieve a definite advantage as its scale increases**. Due to constraints on time and computational resources, we will supplement our study with experiments on LLaVA-Next-34B in future work. However, we posit that our methodology is decoupled from the model’s parameter count, and thus can be effectively applied to models of any scale.
>
> |    Methods     | RefCOCO val | RefCOCO testA | RefCOCO testB | RefCOCO+ val | RefCOCO+ testA | RefCOCO+ testB | RefCOCOg val | RefCOCOg test | LLaVA^(W) | MMHalBench Score ↑ | MMHalBench HalRate ↓ | POPE |
> |---------|-----|-------|-------|-----|-------|-------|-----|------|----------|---------|-----------|------|
> | LLaVA-Next-13B | 87.3 | 91.9 | 81.0 | 81.5 | 88.6 | 72.0 | 83.7 | 84.3 | 71.8 | 2.97 | 0.46 | 82.4 |
> | **+ PerPO** | **89.6** | **93.7** | **85.2** | **82.6** | **90.2** | **75.6** | **85.9** | **86.2** | **72.5** | **3.29** | **0.39** | **83.5** |

---

> ### Author Response · Authors · 2024-12-03
> **Kind Reminder to Reviewer oqhZ**
>
> Dear Reviewer oqhZ,
>
> We sincerely thank you for your thorough review and the professional insights you have provided. We value your feedback greatly, and we hope that our responses have effectively addressed your concerns.
>
> As the discussion period is nearing its end, we would like to remind you that if there are any remaining issues requiring clarification or resolution, please do not hesitate to reach out. We are committed to addressing any questions you may have.
>
> Should you find that our responses have satisfactorily resolved your concerns, we would be truly grateful if you could consider a higher rating, as it is crucial to the final evaluation of our submission. Thank you once again for your time and dedication.
>
> Best,
> Authors

---

### Official Review · Reviewer_NbRT · 2024-11-03

**Soundness:** 3
**Presentation:** 3
**Contribution:** 3
**Rating:** 6
**Confidence:** 3

**Summary:**

The paper presents an approach termed Perceptual Preference Optimization (PerPO) aimed at addressing visual discrimination challenges in generative pre-trained multimodal large language models (MLLMs). The goal of PerPO is to enhance the models' visual discrimination capabilities while maintaining their generative strengths. The approach utilizes discriminative rewarding and listwise preference optimization to align MLLMs with human visual perception processes. The experiments demonstrate that PerPO significantly improves model performance across various visual tasks and contributes to more perceptually aligned and versatile MLLMs.

**Strengths:**

The paper has several strengths. Firstly, it addresses an important challenge in the field of MLLMs, i.e., visual discrimination. Secondly, it presents a novel approach (PerPO) that bridges the gap between generative and discriminative functionalities of MLLMs, and does so effectively according to comprehensive empirical evaluations presented in the paper. Moreover, the PerPO framework reduces the reliance on human annotations for model training, which is a substantial contribution towards scalable and efficient training of MLLMs. Furthermore, the premise of the paper, the methodology, and the evaluations are clearly articulated and well organized, which enhances its readability and digestibility.

**Weaknesses:**

One significant weakness is that the effectiveness of PerPO highly depends on specific datasets, potentially limiting its generalization. The paper also acknowledges a limitation where complex tasks may still require human annotations, which might not always be feasible. Moreover, there might be a lack of comprehensive experiments demonstrating the performance of PerPO across a spectrum of different domains, which would validate its claim of general applicability further.

**Questions:**

1. Could you provide a more detailed explanation of the specific enhancements in MLLM's visual discrimination capabilities by PerPO?
2. How does the model perform when presented with complex visual tasks that require intricate discrimination capabilities?
3. How is the limitation associated with dependency on specific datasets planned to be addressed in future work?
4. Are there plans to integrate PerPO with other existing models or approaches to further improve the overall performance of MLLMs in diverse domains?

---

> ### Author Response · Authors · 2024-11-22
> **Response to Reviewer NbRT**
>
> We sincerely appreciate your careful and constructive reviews. We have responded to each of your comments and questions as follows:
>
> ***Q1: Could you provide a more detailed explanation of the specific enhancements in MLLM's visual discrimination capabilities by PerPO?***
>
> To address your inquiries, we conducted a more detailed analysis of PerPO's impact on both specialized and general capabilities:
>
> 1. **OCR Performance Analysis**: We categorized predictions on the dense OCR test set based on their edit distance from ground truth (GT): 0-0.5, 0.5-0.75, and 0.75-1.0. As presented in the table below, with PerPO, predictions with edit distances <0.5 increased from 18% to 23%, while those >0.75 dropped from 24% to 15%. This indicates that PerPO primarily reduces errors in the worst-performing cases, enhancing overall OCR performance by **effectively constraining the poorest predictions**.
>
>     |  | 0-0.5 |  0.5-0.75 |  0.75-1.0 |
>     |---|:--|---|---|
>     | without PerPO |  18% | 58% | 24% |
>     | with PerPO | 23% | 62% | 15% |
>
> 1. **General Capability Assessment**: We compared models before and after PerPO alignment using GPT-4o and human judgments across three dimensions: response accuracy (RA), instruction adherence (IA), and hallucination reduction (HaR). Results indicate that PerPO mainly facilitates improvements in **response accuracy and hallucination reduction**. This demonstrates that our perceptual alignment encourages models to focus more on image content, effectively mitigating hallucinations stemming from image-unconditional responses.
>
>     |  | RA |  IA |  HaR |
>     |---|:--|---|---|
>     | Win rate as judged by GPT-4o |  80% | 51% | 66% |
>     | Win rate as judged by human users | 84% | 53% | 70% |
>
> We plan to include additional visualizations in the appendix to further elucidate PerPO's mechanisms and validate its effectiveness. We trust these efforts will address your concerns comprehensively.
>
> ***Q2: How does the model perform when presented with complex visual tasks that require intricate discrimination capabilities?***
>
> To evaluate the model's performance on complex visual tasks, we expanded our testing to include more general visual task benchmarks such as **VQAv2, MM-Bench, MM-Vet, and MMMU**. The additional test results are presented in the table below. Evidently, our method demonstrates strong performance on both general and complex visual tasks.
>
> | Methods | MM-Vet |MM-Bench| MMMU  | VQAv2 | LLaVA^(W) |
> |---|--|--|--|---|---|
> | LLaVA-v1.5-7B |  32.9 | 62.3 | 35.7 | 78.5  | 61.8 |
> | + SFT | 31.0 | 62.5 | 36.7 |  78.6  | 62.0 |
> | + DPO | 31.2 | 62.3 | 36.0 | 78.4  | 61.3 |
> | + PerPO | **33.3** | **62.8** | **37.0** | **78.8**  | **64.0** |
>
> Among them, MM-Vet is a widely used metric. It comprehensively assesses model performance across six domains: **recognition, OCR, knowledge, language generation, spatial awareness, and mathematical reasoning**. The following table details a comparison across various sub-tasks. Our method excels across multiple tasks, indirectly suggesting an enhancement in the model's perceptual capabilities.
>
> | Methods | Rec | Ocr| Know | Gen | Spat | Math |   Overall |
> |---|--|--|--|---|---|---|---|
> | LLaVA-v1.5-7B | 44.9  | 26.7 | **22.9**  | 21.5 | 25.6  |  7.7 | 32.9 |
> | + SFT | 43.8 | 25.6 | 16.7 | 20.6  |  24.9 |  7.7 | 31.0 |
> | + DPO | 43.5 | 24.6 | 19.5| 22.5 | 24.5 | 7.7 | 31.2 |
> | + PerPO | **45.1** | **29.3** | 19.5 | **23.0** | **26.8** | **12.7**  | **33.3** |

---

> > ### Author Response · Authors · 2024-11-22
> > **Response to Reviewer NbRT**
> >
> > ***Q3: How is the limitation associated with dependency on specific datasets planned to be addressed in future work?***
> >
> > In this work, we primarily focus on widely-studied visual discrimination tasks with well-defined answers. However, to align with a broader range of visual scenarios in the future, we intend to integrate multiple expert models for data construction and incorporate diverse metrics as our discriminative rewards.
> >
> > Specifically, we can emulate approaches such as PaLI-X [1], Qwen-VL [2], InternVL [3], and LLaMA3.2-Vision [4] by using expert-level OCR models and open-vocabulary detectors to first generate detailed and fine-grained pseudo-labels. Subsequently, we can develop a **composite discriminative reward** centered around algorithms such as IoU, edit distance, and object matching to evaluate the consistency of model outputs against expert pseudo-labels. Finally, leveraging this reward, we can sample diverse preference data to achieve perceptual alignment in complex visual tasks.
> >
> > **References**
> > [1] Chen et al., PaLi-X: On scaling up a multilingual vision and language model.
> > [2] Bai et al., Qwen-vl: A frontier large vision-language model with versatile abilities.
> > [3] Chen et al., Internvl: Scaling up vision foundation models and aligning for generic visual-linguistic tasks.
> > [4] Meta AI, Llama 3.2: Revolutionizing edge AI and vision with open, customizable models.
> >
> > ***Q4: Are there plans to integrate PerPO with other existing models or approaches to further improve the overall performance of MLLMs in diverse domains?***
> >
> > Thank you for noting that. Yes, we do have some exciting plans and we’re thrilled to share them!
> >
> > - **Plan One**: As mentioned in **Q3**, we aim to enhance the generalizability of PerPO by **integrating multiple expert models**, such as OCR, open vocabulary detectors, scene graph detectors, and object trackers. This integration will allow us to develop a versatile perceptual task that combines specialized recognition capabilities. By utilizing IoU, edit distance, and multi-object matching metrics, we intend to create a hybrid discriminative reward system for achieving PerPO.
> >
> > - **Plan Two**: Although our current rule-based reward has improved both specialized and general perceptual capacities and somewhat aligns with human preferences, it’s not enough due to the complexity of human values and ethical considerations. We’re planning to develop a mixed preference optimization method to address these challenges. In the future, we aim to **incorporate multi-objective alignment methods** to ensure our system aligns with human values across various dimensions, such as perception, helpfulness, honesty, and harmlessness.
> >
> > We believe these plans will significantly advance our efforts and are excited about the opportunities that lie ahead.

---

> > > ### Comment · Reviewer_NbRT · 2024-11-28
> > >
> > > Thank you for the authors' detailed response and additional analyses. While the updates strengthen the paper, considering the level of novelty in the work, I believe the paper is marginally above the acceptance threshold and will keep my score at 6.

---

### Official Review · Reviewer_nist · 2024-11-04

**Soundness:** 3
**Presentation:** 3
**Contribution:** 3
**Rating:** 6
**Confidence:** 3

**Summary:**

This work proposes Perceptual Preference Optimization (PerPO) for aligning multimodal large language models (MLLMs) in discriminative vision-language tasks such as object grounding and OCR. By introducing the discriminative reward (measured by the discrepancy between the prediction and the ground truth) into the pairwise weighting in Listwise Preference Optimization (LiPO), PerPO enables sample-efficient alignment of generative MLLMs in discriminative tasks. Empirical results validates that PerPO achieves better performance than DPO and SFT in visual discriminative tasks such as object grounding and dense OCR.

**Strengths:**

1. The writing is clear and smooth, providing great explanation of the background and the proposed method.

2. Compared with DPO and SFT, PerPO enables more sample-efficient alignment for visual discriminative tasks.

3. PerPO also improves general image understanding and mitigates image-unconditional reward hacking.

**Weaknesses:**

1. The evaluation of general vision-language comprehension is based on LLaVA-Bench-in-the-Wild (LLaVA$^W$), a very tiny benchmark with less than 100 samples. The scores may not sufficiently reflect the MLLM image understanding ability. Larger, widely adopted benchmarks such as VQAv2 and MM-Bench are preferred.

2. In Section 5.1, it is claimed that "discriminative reward also aligns well with human," but the results are evaluated by GPT-4o, not human users.

3. From Figure 2(b), the performance on RefCOCO+ seems to saturate after data size > 5k, and more such data can hurt general vision-language capabilities (shown by lowered LLaVA$^W$ scores). Therefore, including more finetuning data does not seem to support PerPO in this scenario. Do SFT and DPO show similar trends? Is PerPO only better than SFT and DPO when the finetuning data scale is small?

4. The tested MLLM and visual perception tasks are somewhat limited (only LLaVA-1.5 and LLaVA-NeXT on object grounding and dense OCR). Thus, the generalizability of PerPO may be unclear. It would be better to include results on more MLLMs (e.g., Cambrian-1, LLaVA-OneVision, Qwen2-VL) and visual tasks (e.g., LISA or GLaMM on segmentation).

**Questions:**

Please check the weakness section above.

---

> ### Author Response · Authors · 2024-11-23
> **Response to Reviewer nist**
>
> We sincerely appreciate your careful and constructive reviews. We have responded to each of your comments and questions as follows:
>
> ***Q1: The evaluation of general vision-language comprehension is based on LLaVA-Bench-in-the-Wild (LLaVA^(W)), a very tiny benchmark with less than 100 samples. The scores may not sufficiently reflect the MLLM image understanding ability. Larger, widely adopted benchmarks such as VQAv2 and MM-Bench are preferred.***
>
> Thank you for your thorough review and valuable suggestions, which have been instrumental in enhancing our manuscript. According to your feedback, we have conducted additional tests on **VQAv2, MM-Bench, MM-Vet, and MMMU** to evaluate the model's general visual capabilities. As presented in the table below, **our PerPO method consistently surpasses SFT and DPO across these metrics**. This further demonstrates that the general visual perceptual capabilities of MLLMs can be improved following PerPO training.
>
> | Methods | MM-Vet |MM-Bench| MMMU  | VQAv2 | LLaVA^(W) |
> |---|--|--|--|---|---|
> | LLaVA-v1.5-7B |  32.9 | 62.3 | 35.7 | 78.5  | 61.8 |
> | + SFT | 31.0 | 62.5 | 36.7 |  78.6  | 62.0 |
> | + DPO | 31.2 | 62.3 | 36.0 | 78.4  | 61.3 |
> | **+ PerPO** | **33.3** | **62.8** | **37.0** | **78.8**  | **64.0** |
>
>
> ***Q2: In Section 5.1, it is claimed that "discriminative reward also aligns well with human," but the results are evaluated by GPT-4o, not human users.***
>
> Thank you for pointing out this issue. We regret the omission of human user assessment in our prior work. Given the high consistency between GPT-4o and human judgments, we previously only employed GPT-4o directly to represent human preferences [1, 2, 3]. **To address any potential discrepancies between GPT-4o and human users, we supplemented our analysis with human evaluations.** For the selected 500 multimodal questions, we invited 20 experts and scholars specializing in computer vision, natural language processing, and human-computer interaction to provide independent assessments.
>
> Specifically, for each question, we calculated average scores across three dimensions: response accuracy, instruction adherence, and hallucination reduction. The winning response was determined based on the magnitude of these average scores. Finally, we synthesized the evaluations from all 20 expert assessors and calculated PerPO’s ultimate win rate by aggregating their judgments. **The evaluation results of GPT-4o and human users are presented as follows. Obviously, the results are similar, with our method achieving over 50% win rates across all test sets.**
>
> |  | LLaVA^(W) |  RefCOCO |  Page-ocr |
> |---|:--|---|---|
> | Win rate as judged by GPT-4o |  56% | 72% | 71% |
> | Win rate as judged by human users | 59% | 76% | 71% |
>
> **References**
> [1] Wu et al., GPT-4o: Visual perception performance of multimodal large language models in piglet activity understanding.
> [2] Huang et al., Human-AI Collaboration Supporting GPT-4o Achieving Human-Level User Feedback in Emotional Support Conversations: Integrative Modeling and Prompt Engineering Approaches.
> [3] Kosaka et al., When ChatGPT-4o Is (Less) Human-Like: Preliminary Subjective Rating Tests for Psycholinguistic Research.

---

> ### Author Response · Authors · 2024-11-23
> **Response to Reviewer nist**
>
> ***Q3: From Figure 2(b), the performance on RefCOCO+ seems to saturate after data size > 5k, and more such data can hurt general vision-language capabilities (shown by lowered LLaVA^(W) scores). Therefore, including more finetuning data does not seem to support PerPO in this scenario. Do SFT and DPO show similar trends? Is PerPO only better than SFT and DPO when the finetuning data scale is small?***
>
> Thank you for your thorough review and for raising professional questions. In the following text, we analyze the reason behind this phenomenon and provide additional experiments using SFT and DPO. We hope our response will meet with your approval.
>
> - **The reason for this phenomenon:** As you mentioned, when the quantity of object grounding data significantly increases, the model's specialized ability improves, but its generalization ability decreases, which seems natural. **Further training with extensive specialized data can severely shift the model distribution, leading to the forgetting of pre-training knowledge and a decline in performance on the LLaVA^(W).** We believe that in this case, the general capabilities of the model also decreases under SFT or DPO training. Our approach, however, ensures that the model achieves excellent performance in both specialized and general capabilities with an appropriate amount of high-quality aligned data.
> - **The performance of SFT and DPO:** To investigate whether DPO and SFT exhibit similar performance trends, we conducted additional experiments using datasets across different scales (3k, 5k, 7k, 9k, and 11k). For a fair comparison with PerPO, we also evaluated on LLaVA^(W) and RefCOCO+. The specific experimental results are presented in the tables below. From these, we can derive two insights.
>    1. As the data size increases, SFT and DPO models exhibit a similar trend. They also forget previously learned knowledge and decline in general capability.
>    2. PerPO outperforms both DPO and SFT across general and specialized capabilities, regardless of fine-tuning data scale.
>
> | Methods | LLaVA^(W) (3k)| LLaVA^(W) (5k)| LLaVA^(W) (7k)| LLaVA^(W) (9k)| LLaVA^(W) (11k)|
> |---------|-----------|-----|-----|-----|-----|
> | SFT     | 62.2      | 61.3| 61.6| 60.2| 60.9|
> | DPO     | 62.4      | 61.3| 62.2| 60.6| 60.7|
> | PerPO   | **65.2**  | **61.6**| **62.6**| **60.7**| **61.1**|
>
> | Methods|  RefCOCO+ (3k)| RefCOCO+ (5k)| RefCOCO+ (7k) | RefCOCO+ (9k) | RefCOCO+ (11k) |
> |---------|-----------|-----|-----|-----|-----|
> | SFT     |53.0      | 55.9| 55.4| 55.3| 56.1|
> | DPO     |53.2      | 55.1| 54.3| 54.5| 55.6|
> | PerPO   |**56.4**| **57.5**| **57.2**| **57.4**| **57.5**|
>
> ***Q4: The tested MLLM and visual perception tasks are somewhat limited (only LLaVA-1.5 and LLaVA-NeXT on object grounding and dense OCR). Thus, the generalizability of PerPO may be unclear. It would be better to include results on more MLLMs (e.g., Cambrian-1, LLaVA-OneVision, Qwen2-VL) and visual tasks (e.g., LISA or GLaMM on segmentation).***
>
> We sincerely appreciate your valuable suggestions. To validate the generalization capability of our method, we conducted experiments on other MLLMs for object detection, including **LLaVA-OneVision [1] and Vary-toy [2]**. The experimental results based on LLaVA-OneVision are presented in the table below, demonstrating that the PerPO method achieves a significant advantage over other comparative methods. Due to time constraints, the evaluation of Vary-toy is still ongoing, and we will update the experimental results in due course.
>
>
> |    Methods     | RefCOCO (val) | RefCOCO (testA) | RefCOCO (testB) | RefCOCO+ (val) | RefCOCO+ (testA) | RefCOCO+ (testB) | RefCOCOg (val) | RefCOCOg (test) | LLaVA^(W) | MMHalBench (Score ↑) | MMHalBench (HalRate ↓) | POPE |
> |---------|-----|-------|-------|-----|-------|-------|-----|------|----------|---------|-----------|------|
> | LLaVA-OneVision | 73.6  |  82.6 |  63.8 |  69.4 |  79.5 | 58.2  | 71.1  | 70.8  | 79.7  |  2.70 | 0.41  |  88.3 |
> | + SFT | 74.7  |  83.7 |  65.4 | 70.3  |  80.8 |  59.1 | 72.1  | 71.7  | 77.9  |  2.73 |  0.40 | 88.1  |
> | + DPO| 79.5  |  86.5 |  71.1 | 74.6  |  83.4 | 64.5  | 76.3  |  76.1 | 80.1  | 2.75  |  0.39 | 88.4  |
> | **+ PerPO** | **82.2**  |  **88.1** |  **75.6** | **77.3**  | **85.3**  | **68.4**  | **79.6**  | **79.9**  | **83.3**  | **2.82**  | **0.37**  | **88.8**  |
>
> Although our method has been validated solely on object detection and dense OCR tasks, it is theoretically applicable to other discriminative visual tasks. We regret that we cannot provide quantitative results for diverse tasks due to time and computational constraints. Nevertheless, as highlighted in the Limitations and Future Work section, we plan to explore its applicability to broader tasks in the future.
>
> **References**
> [1] Li et al., Llava-onevision: Easy visual task transfer.
> [2] Wei et al., Small language model meets with reinforced vision vocabulary.

---

> > ### Comment · Reviewer_nist · 2024-11-25
> >
> > Thank you very much for the detailed response and updates. My previous concerns have been addressed.

---

### Official Review · Reviewer_pPGQ · 2024-11-04

**Soundness:** 1
**Presentation:** 2
**Contribution:** 2
**Rating:** 3
**Confidence:** 4

**Summary:**

This paper proposes PerPO, a listwise reward optimization algorithm to address the poor visual discrimination performance of MLLMs and the reward hacking issues in DPO. Specifically, the authors propose to (1) directly use the task-specific score in the output space as the discriminative reward; (2) using a list of negative discrimination rewards can reduce reward hacking. While the authors show a variety of empirical results, this paper lacks a more informative explanation on why such a method can address the critical issue it mentioned.

**Strengths:**

+ This paper discusses an important topic by investigating the reward optimization issue in MLLMs, and they believe this fundamental issue derives from a misspecified reward definition and reward hacking in DPO. They propose corresponding measures to address issue.
+ This paper contains many empirical results.

**Weaknesses:**

+ While this paper provides many empirical results (e.g. the authors constantly referring to fig.1), the authors only explain their method with intuition, while a more in-depth or theoretical analysis is expected for a paper like this. Only empirical results are not convincing enough.
+ Since this paper simply uses the scalar task-specific discriminative score as the reward, why don't the authors compare their method to PPO while using the reward they defined? It seems equation (8) also seeks to optimize a list of accumulated scalar rewards like PPO did. Could the authors provide more thorough theoretical analysis on the difference of PerPO and PPO, especially in terms of optimization objective and smoothness?
+ It is not very clear why the authors use a list of "negative" rewards? Why woul negative only rewards be sufficient and good enough to optimize using their objective?

**Questions:**

See above.

---

> ### Author Response · Authors · 2024-11-22
> **Response to Reviewer pPGQ**
>
> We sincerely appreciate your careful and constructive reviews. We have responded to each of your comments and questions as follows:
>
> ***Q1: While this paper provides many empirical results (e.g. the authors constantly referring to fig.1), the authors only explain their method with intuition, while a more in-depth or theoretical analysis is expected for a paper like this. Only empirical results are not convincing enough.***
>
> Thanks for your comments. Indeed, we have conducted a theoretical and in-depth analysis in the manuscript. And our theory is also built upon existing methods such as DPO and LiPO. Figure 1 is referenced to offer more robust support through experimental results. The more detailed explanation is given below.
>
> - In Section 3, we have provided an in-depth analysis of the method from theoretical perspective. As mentioned from line 216 to line 241, we argue that **the theoretical foundation underlying PerPO as a preference optimization method is its equivalence to Empirical Risk Minimization (ERM)**, which is widely regarded as highly effective in traditional computer vision. Simultaneously, from a practical standpoint, we substantiate the effectiveness of using discriminative rewards for perceptual preference optimization, as detailed in Section 5.
> - Notably, **the core of our method is to design a reward function for the perception of MLLMs**, which is subsequently optimized using DPO and LiPO methods. **Our theoretical framework is also built upon these established principles.**
>
> ***Q2: Since this paper simply uses the scalar task-specific discriminative score as the reward, why don't the authors compare their method to PPO while using the reward they defined? It seems equation (8) also seeks to optimize a list of accumulated scalar rewards like PPO did. Could the authors provide more thorough theoretical analysis on the difference of PerPO and PPO, especially in terms of optimization objective and smoothness?***
>
> Regarding your concerns, we provide explanations from the following perspectives. We hope this clarification meets with your approval.
>
> 1. **Comparing with PPO may be unnecessary:** The discriminative reward designed in our method can indeed be used for PPO training; however, this is not our primary focus. In fact, the key contribution of our paper lies in proposing a novel discriminative reward to enhance MLLMs' perceptual performance and employing a direct preference optimization method for training.
>
> 2. **Comparing with PPO is unfair:** PPO optimizes using environmental reward signals via policy gradients and value function estimation. Though broadly effective, it often requires substantial resources and tuning for complex tasks. DPO is more widely applied in MLLMs [1, 2, 3], employing direct optimization of user or system preferences to refine the policy. Compared to PPO, DPO offers certain effectiveness and stability while significantly conserving resources [4]. Our approach, based on direct preference optimization, differs significantly from the proximal policy optimization, making an equivalent comparison unfeasible.
>
> 3. **The essence of PerPO's optimization objective:** Equation (8) demonstrates that with our proposed reward, the optimization objective is equivalent to Empirical Risk Minimization (ERM). In other words, the essence of PerPO is to optimize discriminative rewards, which is not the same as PPO to maximize reward expectations through iterative strategy updates. This also  theoretically explains why our method enhances the discriminative capabilities of MLLMs.
>
> **References**
> [1] Wang et al., mDPO: Conditional Preference Optimization for Multimodal Large Language Models.
> [2] Liu et al., Mia-dpo: Multi-image augmented direct preference optimization for large vision-language models.
> [3] Zhu et al., Self-supervised visual preference alignment.
> [4] Ivison et al., Unpacking DPO and PPO: Disentangling Best Practices for Learning from Preference Feedback.

---

> > ### Author Response · Authors · 2024-11-22
> > **Response to Reviewer pPGQ**
> >
> > ***Q3: It is not very clear why the authors use a list of "negative" rewards? Why would negative only rewards be sufficient and good enough to optimize using their objective?***
> >
> > Thank you for pointing out this issue. We apologize for any misunderstanding that may have arisen from our expression. A thorough review of the manuscript revealed two distinct applications of the term `negative`: in the context of rewards and samples. We offer the following elucidation of these concepts.
> > - **Reward is the negative value of errors:** The reward we use is not a traditional negative reward. Instead, the use of negative is intended to convey its inverse relationship with errors. Essentially, our reward is the negative value of errors. In response to this point, we will modify the expression in the original manuscript.
> > - **Negative samples come from the model's responses:** In our paper, negative samples denote model-generated responses that deviate from the ground truth. That is to say, the term "negative" here is relative to the discriminative ground truth. Utilizing model-generated responses for alignment training to unlock the model’s potential is a widely acknowledged effective approach. This explains our assertion that leveraging listwise negative samples can enhance MLLMs’ visual comprehension abilities. We will provide a detailed explanation of the concept of negative samples in the manuscript.

---

> > > ### Comment · Reviewer_pPGQ · 2024-11-26
> > >
> > > Thank the authors for their response! Given the author's response to Q2, the main novelty for PerPo is the discriminative reward instead of the preference optimization algorithm, which is a direct application of LiPO. While I acknowledge the insight for the discriminative reward, it seems to be a bit trivial and not very convincing whether it will be effective and robust in broader and more complex tasks.

---

> > > > ### Author Response · Authors · 2024-12-03
> > > > **Follow-up the response**
> > > >
> > > > We greatly appreciate your response. In the following, we will provide detailed clarifications addressing the concerns you have raised. We hope this will help resolve any uncertainties and provide a better understanding of our approach.
> > > >
> > > > 1. **Our approach is not trivial; rather, we view it as simple yet effective.** By employing listwise preference data for model post-training, we significantly enhance the model’s perceptual capabilities. These preference data, derived from discriminative evaluation criteria, are easily scalable for large-scale augmentation. **Crucially, our method holds substantial significance for the multimodal academic community.** It effectively addresses the reward hacking phenomenon prevalent in previous approaches, marking a notable advancement in perceptual alignment.
> > > > 2. **Our approach is grounded in discriminative rewards, yet extends beyond just rewards.** Additional modeling methods and detailed analyses are key contributions, offering valuable insights. For example, the alignment stage of MLLMs should prioritize perceptual alignment, as pairwise preference alignment based solely on language models is insufficient. In other words, models must have a deeper understanding of image content for more accurate responses.
> > > > 3. **In this paper, we concentrate on OCR and object grounding scenarios, not due to an inability to generalize to more complex contexts, but because these widely recognized and practically valuable applications effectively showcase the utility of our method.** Future work will extend to more intricate scenarios using hybrid multi-expert model labeling and mixed reward strategies.

---

### Author Response · Authors · 2024-11-27
**Common Response**

We are grateful to the reviewers for their comprehensive review and valuable insights. Your thoughtful comments have significantly contributed to improving the quality of our manuscript. Here, we have compiled a summary of the revisions to clearly outline the updates made to the original manuscript. Furthermore, given the similarity of some comments and questions, we have summarized common issues and provided a general response.

1. **Manuscript updates.** We have listed the modifications made to the original manuscript. For detailed content, please refer to the revised version, where the changes are highlighted in blue font.
    1. Add the introduction of "negative samples" in lines 73-76.
    2. Revise the expression related to "negative reward" in lines 51-53 and 157-158.
    3. Replace the GPT-4o judgment results in Fig.3 with human users' assessment and revise the content in lines 416-421.
    4. Add the experimental results for PerPO’s generalization assessment in Appendix A.1.
    5. Add the performance of PerPO on general vision tasks in Appendix A.2.
    6. Add the assessment details of PerPO by GPT-4o and human users in Appendix A.3.

2. **Common issues.** We have summarized and addressed the common concerns raised by the reviewers as follows.
    1. **PerPO's Performance in Complex Visual Tasks.** Despite being trained on specific discriminative tasks, PerPO’s enhanced perceptual capabilities have resulted in notable improvements across complex visual tasks. To substantiate this claim, we conducted additional evaluations using more general visual benchmarks, including **MM-Vet [1], MM-Bench [2], MMMU [3], and VQAv2 [4]**.
    2. **PerPO's Generalization Capability.** To validate our method’s generalizability, we conducted additional comparative experiments utilizing the **LLaVA-OneVision [5]** framework. Moreover, we extended our investigation to encompass **LLaVA-Next-13B [6]**, aiming to assess the efficacy of discriminative rewards in scaling up the model. Notably, PerPO consistently exhibited superior performance across all experimental settings when compared to alternative methods.
    3. **Future Work Plan for PerPO.** To align with a broader range of visual scenarios in the future, we intend to **integrate multiple expert models for data construction and incorporate diverse metrics as our discriminative rewards**. Leveraging the composite discriminative reward, we can sample diverse preference data to achieve perceptual alignment in complex visual tasks.

We genuinely hope our summary offers meaningful insights. We have thoughtfully addressed each comment from every reviewer and look forward to your support and approval.

**References**
[1] Yu et al., Mm-vet: Evaluating large multimodal models for integrated capabilities.
[2] Liu et al., Mmbench: Is your multi-modal model an all-around player?
[3] Yue et al., Mmmu: A massive multi-discipline multimodal understanding and reasoning benchmark for expert agi.
[4] Goyal et al., Making the v in vqa matter: Elevating the role of image understanding in visual question answering.
[5] Li et al., Llava-onevision: Easy visual task transfer.
[6] Liu et al., LLaVA-NeXT: Improved reasoning, OCR, and world knowledge.

---

### Meta-Review · Area_Chair_H4u2 · 2024-12-20

**Metareview:**

The paper addresses visual discrimination in multimodal large language models (MLLMs) by introducing Perceptual Preference Optimization (PerPO), a method that extends listwise preference optimization with discriminative rewards based on the discrepancy between predictions and ground truth. A key innovation lies in the use of a list of negative discrimination rewards to counter reward hacking issues commonly encountered in Direct Preference Optimization (DPO). This design enables sample-efficient alignment of generative MLLMs with human perception, improving performance in visual tasks such as object grounding and OCR. Claimed advantages over prior approaches include enhanced task-specific visual discrimination and robust generalization across models and tasks. Empirical results validate PerPO’s superiority over DPO and SFT.

Four reviewers evaluated this paper, highlighting its major strengths as its ability to bridge the gap between generative and discriminative functionalities of MLLMs, its significant contributions to visual discrimination tasks, its superiority over baseline methods such as DPO and SFT, and the framework's reduced reliance on human annotations.

However, reviewers raised several concerns about the paper, particularly its reliance on empirical results and intuition without sufficient theoretical analysis. The distinction between PerPO and PPO, specifically regarding optimization objectives and smoothness, was highlighted as an area needing deeper exploration. Questions were also raised about the use of negative-only rewards, with reviewers questioning their sufficiency and optimality for the proposed objective. Another significant concern was the limited scope of evaluation, which predominantly focused on small benchmarks (e.g., LLaVA-Bench-in-the-Wild) and a narrow range of tasks (e.g., object grounding and OCR), leaving broader generalizability unverified. Furthermore, claims about perceptual alignment being the primary limitation in MLLMs' visual discrimination were not substantiated with sufficient evidence, as alternative factors such as instruction tuning or pre-training dataset limitations were not thoroughly examined.

During the discussion phase, the authors demonstrated PerPO’s performance on complex visual tasks with additional evaluations conducted on general benchmarks such as MM-Vet, MM-Bench, MMMU, and VQAv2, and also validated the method’s generalization capability with experiments performed using LLaVA-OneVision and LLaVA-Next-13B.

The reviewers partially acknowledged the authors' clarifications and additional experimental results. However, Reviewers pPGQ and NbRT expressed ongoing concerns about the novelty of the proposed approach, noting that it primarily represents a direct application of LiPO with a discriminative reward. Furthermore, while additional results on more benchmarks and tasks and with advanced models were provided during the rebuttal phase, a comprehensive and systematic evaluation was not fully presented due to time constraints.

The ACs concur that the lack of comprehensive and systematic evaluation is a legitimate concern and a notable weakness in the current manuscript. Additionally, the novelty of the proposed approach should be more clearly explained, and the claims regarding perceptual alignment as the primary limitation in MLLMs' visual discrimination should be more thoroughly examined. The manuscript also requires substantial rewriting to better organize the new results and, in particular, integrate them into the main paper. Recognizing the validity of the weaknesses raised by the reviewers, the ACs assess that these shortcomings outweigh the current merits of the paper. Consequently, the ACs cannot recommend the acceptance of the current manuscript.

**Additional Comments On Reviewer Discussion:**

The ACs acknowledge the validity of the concerns raised by the reviewers, particularly regarding the incomplete empirical validation and limited novelty, and conclude that these weaknesses outweigh the current strengths of the paper. Further details can be found in the meta-review.

---

### Decision · Program_Chairs · 2025-01-22

Reject